

# Complete mitogenome of *Olidiana ritcheriina* (Hemiptera: Cicadellidae) and phylogeny of Cicadellidae

Xian-Yi Wang[1], Jia-Jia Wang[1], Zhi-Hua Fan[2] and Ren-Huai Dai[1]

[1] Institute of Entomology, Guizhou University, The Provincial Key Laboratory for Agricultural Pest Management of Mountainous Region, Guizhou, Guiyang, China
[2] Jingtanggang Customs House, Tangshan, Hebei, Tangshan, China

## ABSTRACT

**Background**. Coelidiinae, a relatively large subfamily within the family Cicadellidae, includes 129 genera and ∼1,300 species distributed worldwide. However, the mitogenomes of only two species (*Olidiana* sp. and *Taharana fasciana*) in the subfamily Coelidiinae have been assembled. Here, we report the first complete mitogenome assembly of the genus *Olidiana*.

**Methods**. Specimens were collected from Wenxian County (Gansu Province, China) and identified on the basis of their morphology. Mitogenomes were sequenced by next-generation sequencing, following which an NGS template was generated, and this was confirmed using polymerase chain reaction and Sanger sequencing. Phylogenic trees were constructed using maximum likelihood and Bayesian analyses.

**Results**. The mitogenome of *O. ritcheriina* was 15,166 bp long, with an A + T content of 78.0%. Compared with the mitogenome of other Cicadellidae sp., the gene order, gene content, gene size, base composition, and codon usage of protein-coding genes (PCGs) in *O. ritcheriina* were highly conserved. The standard start codon of all PCGs was ATN and stop codon was TAA or TAG; *COII*, *COIII*, and *ND4L* ended with a single T. All tRNA genes showed the typical cloverleaf secondary structure, except for *trnSer*, which did not have the dihydrouridine arm. Furthermore, the secondary structures of rRNAs (*rrnL* and *rrnS*) in *O. ritcheriina* were predicted. Overall, five domains and 42 helices were predicted for *rrnL* (domain III is absent in arthropods), and three structural domains and 27 helices were predicted for *rrnS*. Maximum likelihood and Bayesian analyses indicated that *O. ritcheriina* and other Coelidiinae members were clustered into a clade, indicating the relationships among their subfamilies; the main topology was as follows: (Deltocephalinae + ((Coelidiinae + Iassinae) + ((Typhlocybinae + Cicadellinae) + (Idiocerinae + (Treehopper + Megophthalminae))))). The phylogenetic relationships indicated that the molecular taxonomy of *O. ritcheriina* is consistent with the current morphological classification.

Corresponding author
Ren-Huai Dai, rhdai@gzu.edu.cn

# INTRODUCTION

Coelidiinae is a relatively large subfamily within the Cicadellidae family, and it includes 129 genera and approximately 1,300 species (*Nielson, 2015*), including some species that

serve as vectors of pathogens causing economically important plant diseases (*Du et al., 2017*; *Frazier, 1975*; *Li & Fan, 2017*; *Maramorosch, Harris & Futuyma, 1981*; *Zhang, 1990*). However, the taxonomic status of some species, on the basis of their morphology, remains controversial, and the phylogenetic relationships among major lineages of Membracoidea remain poorly understood (*Dietrich et al., 2017*). Moreover, knowledge regarding the taxonomic status of *Olidiana* within Cicadellidae and its phylogenetic relationship with other leafhopper genera is limited.

Complete mitogenomes provide large and diverse datasets for species delineation, and such mitogenomes have extensively been used for evolutionary studies of insects, particularly members of the orders Lepidoptera, Diptera, and Hemiptera (*Salvato et al., 2008*; *Wang et al., 2011*; *Du et al., 2017*; *Su & Liang, 2018*; *Wang et al., 2018*; *Li et al., 2017*). To date, approximately 35 species (26 complete and nine nearly complete) of the Cicadellidae mitogenome are available in GenBank. However, the mitogenomes of only two species [*Olidiana* sp. (partial genome, KY039119.1) and *Taharana fasciana* (NC_036015.1)] have previously been published for Coelidiinae, the largest subfamily of Cicadellidae.

*Olidiana* McKamey is the largest leafhopper genus in the tribe Coelidiini and it comprises 91 species. Among these, 54 species have been reported from China. However, to date, none of the characterized mitogenomes of the *Olidiana* sp. is complete; this lack of information restricts our understanding of the evolution of the Coelidiinae sp. at the genomic level. Therefore, new mitogenomic data will provide insights for determining the phylogenetic relationships and evolution of Cicadellidae in the future.

*Olidiana ritcheriina*, first described in 1990 (*Zhang, 1990*), is widely distributed throughout the Chinese provinces of Shaanxi, Hubei, Hunan, Guangdong, Hainan, Guangxi, Sichuan, Guizhou, and Yunnan. Therefore, a complete mitogenome of *O. ritcheriina* (GenBank accession NO.: MK738125) was sequenced to elucidate the phylogenetic status and relationships of the Coelidiinae sp.

## MATERIALS & METHODS

### Sample collection and identification

The use of the specimens collected for this study was approved. The specimens were collected from Wenxian County, Gansu Province, China (32°95′N, 104°68′E) on October 17, 2018, and identified on the basis of their morphological characteristics, as described by *Zhang (1990)* and *Li & Fan (2017)*. Fresh specimens were preserved in absolute ethanol and stored at −20 °C until DNA extraction.

### DNA extraction

Genomic DNA was extracted from the whole body of adult males (after removing the abdomen and wing) using DNeasy©Tissue Kit (Qiagen, Hilden, Germany). The samples were incubated at 56 °C for 6 h for completely lysing the cells, and total genomic DNA was eluted in 100 μL of double-distilled water; the remaining steps were performed according to the manufacturer's instructions. The extracted genomic DNA was stored at −20 °C until further use. Voucher specimens with male genitalia and DNA samples have been deposited at the Institute of Entomology, Guizhou University, Guiyang, China.

## Polymerase chain reaction (PCR) amplification and sequencing

Mitogenomes were sequenced using next-generation sequencing (Illumina HiSeq 4000 and 2 Gb raw data; Berry Genomic, Beijing, China), and two sequence fragments were reconfirmed by PCR amplification using primers (Table S1). Following this, an NGS template was generated and this was further confirmed using PCR and Sanger sequencing. PCR amplification of overlapping sequence fragments was performed using universal primers (Table S1). Two pairs of species-specific primers were designed using Primer Premier 6.0 (Premier Biosoft, Palo Alto, CA, USA) to amplify the control region (Table S1). PCR was performed using a PCR master mix (Sangon Biotech Co. Ltd., Shanghai, China), according to the manufacturer's instructions.

## Sequence analysis

Next-generation sequences were assembled using Geneious R9 (*Kearse et al., 2012*). The assembled mitochondrial gene sequences were compared with the homologous sequences of *Olidiana* sp. (KY039119) and *T. fasciana* (KY886913) retrieved from GenBank and identified through BLAST searches in NCBI to confirm sequence accuracy. The sequences obtained by PCR amplification and TA cloning were assembled using SeqMan in the DNAStar software package (DNASTAR, Inc., Madison, WI, USA). The mitogenomes were annotated using the MITOS webserver (*Bernt et al., 2013*). Base composition and relative synonymous codon usage (RSCU) were analyzed using MEGA 6.06 (*Tamura et al., 2013*), and the boundaries and secondary structures of 22 tRNA genes were determined using tRNAscan-SE version 1.21 (*Schattner, Brooks & Lowe, 2005*) and ARWEN version 1.2 (*Laslett & Canbäck, 2008*). rRNA genes were identified on the basis of the locations of adjacent tRNA genes and comparisons with sequences of other Hemipterans. The secondary structures of rRNAs were inferred on the basis of models proposed for other Hemiptera (*Wang, Li & Dai, 2017*; *Su et al., 2018*). Helices were numbered according to the convention established by the Comparative RNA Web Site (*Cannone et al., 2002*). Strand asymmetry was calculated using the following formulas: AT skew = $(A - T)/(A + T)$, GC skew = $(G - C)/(G + C)$ (*Perna & Kocher, 1995*). Intergenic spacers and overlapping regions between genes were manually counted.

## Sequence alignment and phylogenetic analysis

The phylogenetic analysis included complete or nearly complete mitogenome sequences of 42 insect species, namely 35 leafhoppers, 5 treehoppers, 2 froghoppers (*Tettigades auropilosa* and *Cosmoscarta bispecularis*) as outgroups, and *O. ritcheriina*, which was newly sequenced (Table 1).

Each PCG and rRNA sequence was aligned using the MAFFT algorithm in Translator X (*Abascal, Zardoya & Telford, 2010*; *Katoh, Rozewicki & Yamada, 2019*) and MAFFT v7.0 online serve with the G-INS-i strategy (*Castresana, 2000*), respectively. Poorly aligned sites were removed using Gblocks 0.91b (*Castresana, 2000*) under default settings, except that the gap sites were toggled as "none". Subsequently, the resulting 15 alignments were assessed and manually corrected using MEGA 6 (*Tamura et al., 2013*).

The following five datasets were concatenated for phylogenetic analysis: (1) P123: all codon positions of 13 PCGs (10,116 bp); (2) P12: first and second codon positions of 13

**Table 1** Summary of the mitogenomes used in this study.

| | Species | Size (bp) | A+T (%) | Accession number | Reference |
|---|---|---|---|---|---|
| Cicadellinae | *Bothrogonia ferruginea* | 15,262 | 76.5 | KU167550 | Unpublished |
| | *Homalodisca vitripennis* | 15,304 | 78.4 | NC_006899 | Unpublished |
| Coelidiinae | *Olidiana ritcheriina* | 15,166 | 78.0 | MK738125 | **This study** |
| | *Olidiana* sp.[a] | 15,253 | 78.1 | KY039119 | Unpublished |
| | *Taharana fasciana* | 15,161 | 77.9 | KY886913 | *Wang, Li & Dai (2017)* |
| Deltocephalinae | *Agellus* sp.[a] | 14,819 | 75.8 | KX437738 | *Song, Cai & Li (2018)* |
| | *Alobaldia tobae*[a] | 16,026 | 77.3 | KY039116 | *Song, Cai & Li (2017)* |
| | *Cicadula* sp.[a] | 14,929 | 74.1 | KX437724 | *Song, Cai & Li (2018)* |
| | *Drabescoides nuchalis* | 15,309 | 75.6 | NC_028154 | *Wu et al. (2016)* |
| | *Exitianus indicus*[a] | 16,089 | 75.1 | KY039128 | *Song, Cai & Li (2017)* |
| | *Japananus hyalinus* | 15,364 | 76.6 | NC_036298 | *Du et al. (2017)* |
| | *Macrosteles quadrilineatus* | 16,626 | 78.0 | NC_034781 | *Mao, Yang & Bennett (2017)* |
| | *Maiestas dorsalis* | 15,352 | 78.7 | NC_036296 | *Du et al. (2017)* |
| | *Nephotettix cincticeps* | **14,805** | 77.6 | NC_026977 | Unpublished |
| | *Norvellina* sp.[a] | 15,594 | 74.5 | KY039131 | *Song, Cai & Li (2017)* |
| | *Orosius orientalis*[a] | 15,513 | 72.0 | KY039146 | *Song, Cai & Li (2017)* |
| | *Phlogotettix* sp. | 15,136 | 77.9 | KY039135 | *Song, Cai & Li (2017)* |
| | *Scaphoideus maai* | 15,188 | 77.2 | KY817243 | *Du, Dai & Dietrich (2017)* |
| | *Scaphoideus nigrivalveus* | 15,235 | 76.6 | KY817244 | *Du, Dai & Dietrich (2017)* |
| | *Scaphoideus varius* | 15,207 | 75.9 | KY817245 | *Du, Dai & Dietrich (2017)* |
| | *Tambocerus* sp. | 15,955 | 76.4 | KT827824 | *Yu et al. (2017)* |
| | *Yanocephalus yanonis* | 15,623 | **74.6** | NC_036131 | *Song, Cai & Li (2017)* |
| Iassinae | *Trocnadella arisana* | 15,131 | **80.7** | NC_036480 | Unpublished |
| Idiocerinae | *Idioscopus clypealis* | 15,393 | 78.3 | MF784430 | *Dai, Wang & Yang (2018)* |
| | *Idioscopus laurifoliae* | **16,811** | 79.5 | MH433622 | *Wang et al. (2018)* |
| | *Idioscopus sp myrica* | 15,423 | 77.9 | MH492317 | *Wang et al. (2018)* |
| | *Idioscopus nitidulus* | 15,287 | 78.6 | NC_029203 | *Choudhary et al. (2018)* |
| | *Populicerus populi* | 16,494 | 77.2 | MH492318 | *Wang et al. (2018)* |
| Megophthalminae | *Durgades nigropicta* | 15,974 | 78.8 | NC_035684 | *Wang et al. (2017)* |
| | *Japanagallia spinosa* | 15,655 | 76.6 | NC_035685 | *Wang et al. (2017)* |
| Treehopper | *Darthula_hardwickii* | 15,355 | 78.0 | NC_026699 | *Liang, Gao & Zhao (2016)* |
| | *Entylia carinata* | 15,662 | 78.1 | NC_033539 | *Mao, Yang & Bennett (2016)* |
| | *Leptobelus gazella* | 16,007 | 78.8 | NC_023219 | *Zhao & Liang (2016)* |
| | *Leptobelus* sp. | 15,201 | 77.5 | JQ910984 | *Li et al. (2017)* |
| | *Tricentrus* sp. | 15,419 | 78.5 | KY039118 | Unpublished |
| Typhlocybinae | *Empoasca onukii* | 15,167 | 78.3 | NC_037210 | *Liu et al. (2017)* |
| | *Empoasca* sp.[a] | 15,116 | 76.8 | KX437737 | *Song, Cai & Li (2018)* |
| | *Empoasca vitis* | 15,154 | 78.3 | NC_024838 | *Zhou et al. (2016)* |
| | *Illinigina* sp.[a] | 14,803 | 76.0 | KY039129 | *Song, Cai & Li (2017)* |
| | *Typhlocyba* sp. | 15,223 | 77.1 | KY039138 | *Song, Cai & Li (2017)* |
| Cicadoidea | *Tettigades auropilosa* | 14,944 | 75.0 | KM000129 | Unpublished |
| Cercopoidea | *Cosmoscarta bispecularis* | 15,426 | 78.5 | KP064511 | *Yang, Liu & Liang (2016)* |

**Notes.**
[a]Incomplete mitochondrial genomes.

PCGs (6,744 bp); (3) P123-rR: P123 and two rRNAs (11,934 bp); (4) P12-rR: P12 and two rRNAs (8,562 bp); and (5) AA: amino acid sequences of 13 PCGs (3,371 bp). The potential substitution saturation of four datasets (P123, P12, P123-rR, and P12-rR) was assessed using the index of substitution saturation (*Iss*) proposed by *Xia et al. (2003)* and implemented in DAMBE 5 (*Xia, 2013*).

Maximum likelihood (ML) analysis was performed using IQ-TREEv1.6.3 (*Nguyen et al., 2014*) with the best model for each partition selected under the corrected Akaike Information Criterion (AIC) using PartitionFinder2 (Table S2) (*Miller, Pfeiffer & Schwartz, 2010*) and evaluated using the ultrafast bootstrap approximation approach for 10,000 replicates. Bayesian (BI) analysis was performed using MrBayes 3.2 (*Ronquist et al., 2012*). Two independent runs with four simultaneous Markov chains (one cold and three incrementally heated at $T = 0.2$) were run for 50,000,000 generations, sampling every 100 generations under the GTR+I+G model. The best models were then selected on the basis of the corrected AIC (*Nylander et al., 2004*). The phylogenetic trees were visualized using FigTree 1.4.2.

## RESULTS

### General features of the *O. ritcheriina* mitogenome

The complete mitogenome of *O. ritcheriina* (MK738125) was 15,166 bp long, which is within the range of the complete mitogenomes of other Cicadellidae sp. (*Nephotettix cincticeps*, 14,805 bp and *Idioscopus laurifoliae*, 16,811 bp) (Table 1). The mitogenome comprised 37 genes (13 PCGs, 22 tRNAs, and two rRNAs) and a large A + T-rich D-loop control region (Fig. 1). The majority strand (J strand) harbored most of the genes (nine PCGs and 14 tRNAs), whereas the minority strand (N strand) harbored the remaining genes (four PCGs, two rRNAs, and eight tRNAs) (Fig. 1; Table 2). Moreover, the mitogenome of *O. ritcheriina* comprised intergenic spacers of 1 to 12 bp long at nine different loci. A total of 12 gene pairs overlapped with one another, with overlap lengths ranging from 1 to 13 bp. In addition, 16 gene pairs, including *rrnL–trnV* and *trnV–rrnS* (Table 2), were directly adjacent to one another. With a multicopy of *trnI* (AAT) located between the control region and *trnI–trnQ–trnM*, the mitogenome of *O. ritcheriina* exhibited a strong A + T bias. The A + T content of the whole genome was 78.0% (44.6% A, 33.4% T, 8.5% G, and 13.5% C) (Table 3); this percentage was between the A + T content of *Yanocephalus yanonis* (74.6%) and *Trocnadella arisana* (80.7%) (Table 1). The segment with the highest A + T content was present in the control region (83.8%); the A + T content of this segment was generally higher than that of other segments (2 rRNAs, 81.1%; 22 tRNAs, 78.6%; whole genome, 78.0%; and 13 PCGs, 77.2%) (Table 3).

Composition analysis revealed that the mitogenome of *O. ritcheriina* exhibited a positive AT (0.144) and negative GC skew (−0.227) in the whole mitogenome as well as in the 13 PCGs (AT skew: 0.163; GC skew: −0.250), 2 rRNAs (AT skew: 0.160; GC skew: −0.265), and 22 tRNAs (AT skew: 0.111; GC skew: −0.110). However, slightly negative AT (−0.049) and GC (−0.031) skews were detected in the control region (Table 3).

Comparative analysis of the base composition of every component of the mitogenomes of Coelidiinae indicated that the control regions showed the highest A + T content

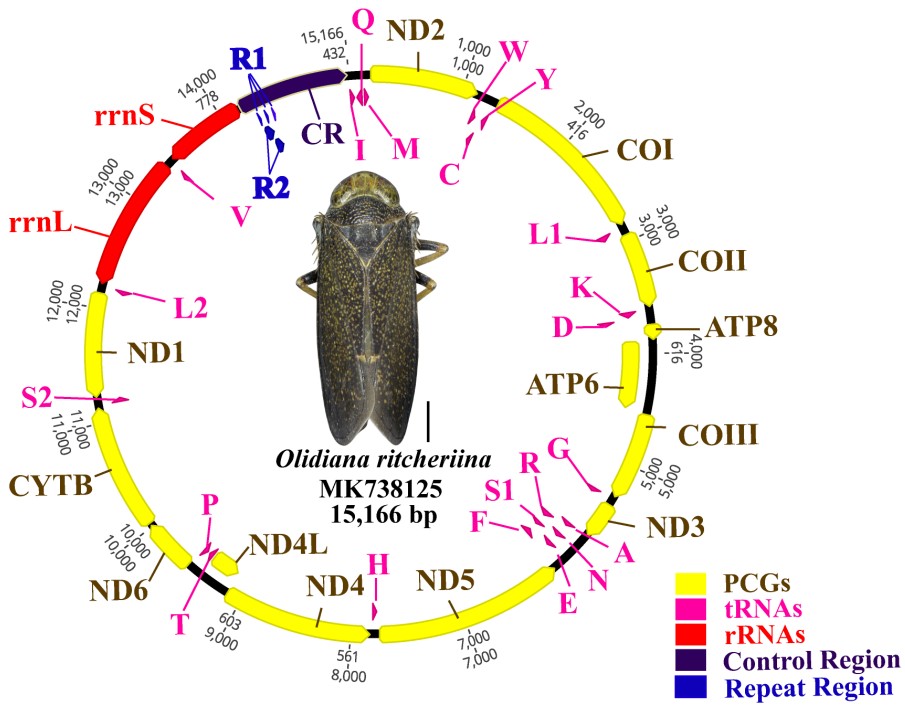

**Figure 1  Circular map of the *Olidiana ritcheriina* mitochondrial genome.** Protein coding and ribosomal genes are shown with standard abbreviations. Transfer RNA (tRNA) genes are indicated using the IUPAC-IUB single letter amino acid codes (L1, CUN; L2, UUR; S2).

(81.4%–83.8%) and that all species exhibited a positive AT (0. 144 to 0.16) or negative GC (−0.227 to −0.23) skew (*Wang, Li & Dai, 2017*) (Table 3).

## PCGs and codon usage

The concatenated lengths of the 13 PCGs of *O. ritcheriina* were 10,116 nucleotide positions. Similar to the mitogenomes of other Cicadellidae sp., *ND5* was the largest gene (1,671 bp) and *ATP8* was the smallest gene (150 bp). Only four PCGs (*ND4*, *ND4L*, *ND5*, and *ND1*) were coded by the minority strand (N strand), whereas the other nine PCGs (*COI*, *COII*, *COIII*, *ATP8*, *ATP6*, *ND2*, *ND3*, *ND6*, and *CYTB*) were coded by the majority strand (J strand). Most PCGs exhibited the typical start codon ATN (ATA/ATT/ATG/ATC) and stop codon TAA or TAG, but *COII*, *COIII*, and *ND4L* showed an incomplete stop codon T.

Analysis of the behavior of PCG codon families revealed an extremely similar codon usage among the mitogenomes of Cicadellidae, with TTA-Leu, ATA-Met, ATT-Ile, and TTT-Phe being the four most frequently used codons (Fig. 2A). Furthermore, the RSCU of *O. ritcheriina* indicated that degenerate codons were biased to use more A/T than G/C at the third codon (Fig. 2B). Similarly, the biased usage of A + T nucleotides was reflected in the codon frequencies.

**Table 2** Composition and skewness of the *Olidiana ritcheriina* mitogenome.

| Gene | Direction | Location | Size (bp) | Start | Stop | Anticodon | Intergenic nucleotides |
|------|-----------|----------|-----------|-------|------|-----------|------------------------|
| *trnI* | J | 1-62 | 62 | – | – | GAT | |
| *trnQ* | N | 64-130 | 67 | – | – | TTG | 1 |
| *trnM* | J | 131-196 | 66 | – | – | CAT | 0 |
| *ND2* | J | 197-1,153 | 957 | ATT | TAA | – | 0 |
| *trnW* | J | 1,152-1,213 | 62 | – | – | TCA | 2 |
| *trnC* | N | 1,201-1,262 | 62 | – | – | GCA | −13 |
| *trnY* | N | 1,263-1,325 | 63 | – | – | GTA | 0 |
| *COI* | J | 1,338-2,873 | 1,536 | ATG | TAA | – | 12 |
| *trnL1(UUR)* | J | 2,874-2,940 | 67 | – | – | TAA | 0 |
| *COII* | J | 2,941-3,616 | 676 | ATT | T | – | 0 |
| *trnK* | J | 3,617-3,687 | 73 | – | – | CTT | 0 |
| *trnD* | J | 3,687-3,748 | 62 | – | – | GTC | −1 |
| *ATP8* | J | 3,750-3,899 | 150 | ATA | TAA | – | 1 |
| *ATP6* | J | 3,893-4,537 | 645 | ATG | TAA | – | −7 |
| *COIII* | J | 4,538-5,315 | 778 | ATG | T | – | 0 |
| *trnG* | J | 5,316-5,375 | 60 | – | – | TCC | 0 |
| *ND3* | J | 5,376-5,729 | 354 | ATA | TAG | – | 0 |
| *trnA* | J | 5,728-5,788 | 61 | – | – | TGC | −2 |
| *trnR* | J | 5,788-5,852 | 65 | – | – | TCG | −1 |
| *trnN* | J | 5,850-5,913 | 64 | – | – | GTT | −3 |
| *trnS1(AGN)* | J | 5,913-5,974 | 62 | – | – | GCT | −1 |
| *trnE* | J | 5,974-6,036 | 63 | – | – | TTC | −1 |
| *trnF* | N | 6,036-6,103 | 68 | – | – | GAA | −1 |
| *ND5* | N | 6,103-7,773 | 1,671 | ATA | TAG | – | −1 |
| *trnH* | N | 7,774-7,834 | 61 | – | – | GTG | 0 |
| *ND4* | N | 7,834-9,150 | 1,317 | ATG | TAG | – | −1 |
| *ND4L* | N | 9,152-9,419 | 278 | ATG | T | – | 1 |
| *trnT* | J | 9,422-9,484 | 63 | – | – | TGT | 1 |
| *trnP* | N | 9,485-9,546 | 62 | – | – | TGG | 0 |
| *ND6* | J | 9,549-10,025 | 477 | ATA | TAA | – | 2 |
| *CYTB* | J | 10,030-11,151 | 1,122 | ATT | TAA | – | 4 |
| *trnS2(UCN)* | J | 11,151-11,214 | 64 | – | – | TGA | -1 |
| *ND1* | N | 11,216-12,146 | 939 | ATT | TAA | – | 1 |
| *trnL2(CUN)* | N | 12,147-12,214 | 68 | – | – | TAG | 0 |
| *rrnL* | N | 12,215-13,394 | 1,180 | – | – | – | 0 |
| *trnV* | N | 13,395-13,454 | 60 | – | – | TAC | 0 |
| *rrnS* | N | 13,455-14,185 | 731 | – | – | – | 0 |
| A+T-rich | | 14,148-14,313 | 166 | – | – | – | 0 |

## tRNAs and rRNAs

All the 22 typical tRNA genes were present in the mitogenome of *O. ritcheriina*, and their lengths ranged between 61 (*trnA* and *trnH*) and 71 bp (*trnK*). All tRNAs were identified

**Table 3 Annotation of the *Olidiana ritcheriina* mitogenome.**

| Regions | Size | A % | G % | T % | C % | A+T % | G+C % | AT skew | GC skew |
|---|---|---|---|---|---|---|---|---|---|
| Whole genome | 15,166 | 44.6 | 8.5 | 33.4 | 13.5 | 78.0 | 22.0 | 0.144 | −0.227 |
| PCGs | 10,890 | 44.7 | 8.8 | 32.1 | 14.4 | 77.2 | 23.2 | 0.163 | −0.250 |
| tRNA genes | 1405 | 43.6 | 9.5 | 34.9 | 11.9 | 78.6 | 21.4 | 0.111 | −0.110 |
| rRNA genes | 1911 | 47.0 | 7.0 | 34.1 | 12.0 | 81.1 | 18.9 | 0.160 | −0.265 |
| Control region | 981 | 39.9 | 7.8 | 43.9 | 8.4 | 83.8 | 16.2 | −0.049 | −0.031 |

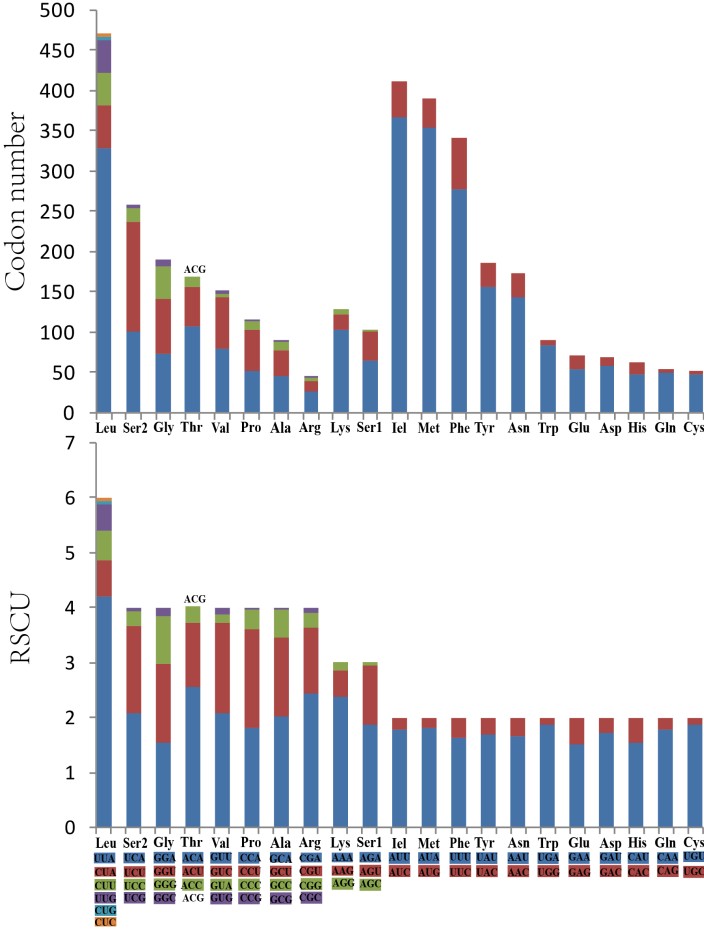

**Figure 2 The codon number and relative synonymous codon usage (RSCU) of PCGs in *Olidiana ritcheriina* mitogenome.**

using tRNAscan-SE (*Schattner, Brooks & Lowe, 2005*) and ARWEN (*Laslett & Canbäck, 2008*). Among these, 14 were located on the J strand and eight on the N strand. All tRNAs exhibited the typical cloverleaf secondary structure, with the exception of *trnS1* (AGN) in

which the dihydrouridine arm formed a loop (Fig. 3). *Abascal et al. (2006)* and *Abascal, Posada & Zardoya (2012)* have shown that the invertebrate mitochondrial genetic code even shifts within the Hemiptera, with *Triatoma* (Cimicomorpha), *Homalodisca* (Cicadellidae), and *Philaenus* (Cercopoidea) using the AGG codon that was translated as Lys instead of Ser; accordingly, our tRNA analysis shows that the AGG codon in *O. ritcheriina* was translated as Lys instead of Ser.

Two rRNA genes (*rrnL* and *rrnS*) in the mitogenomes of Cicadellidae were highly conserved. The putative lengths of the *O. ritcheriina* genes *rrnL* and *rrnS* were 1,180 bp between *trnL2* and *trnV* and 731 bp between *trnV* and the control region, respectively (Tables 2 and 3). In the mitogenomes of Coelidiinae, the length of *rrnL* ranged from 1,178 (*Olidiana* sp.) to 1,192 bp (*T. fasciana*) and that of *rrnS* ranged from 729 (*Olidiana* sp.) to 775 bp (*T. fasciana*). The secondary structure of the *O. ritcheriina* gene *rrnL* comprised five domains (I, II, IV, V, and VI; domain III is absent in arthropods) and 42 helices (Fig. 4). Multiple alignment of the Coelidiinae gene *rrnL* extended over 1,180 positions and comprised 1,016 conserved (86.10%) and 164 variable (13.90%) sites. Domains IV and V were structurally more conserved than the other domains.

The secondary structure of *rrnS* comprised three structural domains and 27 helices (Fig. 5). Multiple alignments of the Coelidiinae gene *rrnS* extended over 730 positions and comprised 586 conserved (80.23%) and 164 variable (19.73%) sites. Domain III was structurally more conserved than domains I and II.

These rRNA secondary structures can be useful for the precise alignment of sequences for phylogenetic studies (*Rijk & Wachter, 1997*). Nevertheless, additional details regarding such rRNA structures should be accumulated in future studies.

## Control region

The control regions (A + T-rich regions) in the mitogenomes of Coelidiinae were not highly conserved, with lengths ranging between 915 (*T. fasciana*) and 1,069 bp (*Olidiana* sp.) and A + T content ranging between 77.9% (*T. fasciana*) and 78.1% (*Olidiana* sp.) (Table 1). The length of the control region of *O. ritcheriina* was 981 bp, with a high A + T content (83.8%) and two repeats: R1 (2 × 49 bp) and R2 (2 × 128 bp) (Fig. 6A). However, the control regions of *T. fasciana* and *Olidiana* sp. comprised a single repeat (Figs. 6B–6C). In addition, the control region of the *O. ritcheriina* showed slightly negative AT (−0.049) and GC (−0.031) skews (Table 3).

## Phylogenetic relationship

Phylogenetic trees were constructed on the basis of five concatenated nucleotide sequence datasets from 40 available mitogenomes of Membracoidea, with two species considered outgroups [Cicadoidea (*T. auropilosa*) and Cercopoidea (*C. bispecularis*)]. Saturation analysis addresses the issue on whether some positions or partitions of a dataset are saturated and to test whether these sites can be used for further phylogenetic analysis. These phylogenetic trees showed uncorrected pairwise divergence in transitions (s) and transversions (v) against divergences calculated with the GTR model, and none of the four candidate nucleotide sequence datasets (Fig. S1A: P123; Fig. S1B: P12; Fig. S1C: P123-rR;

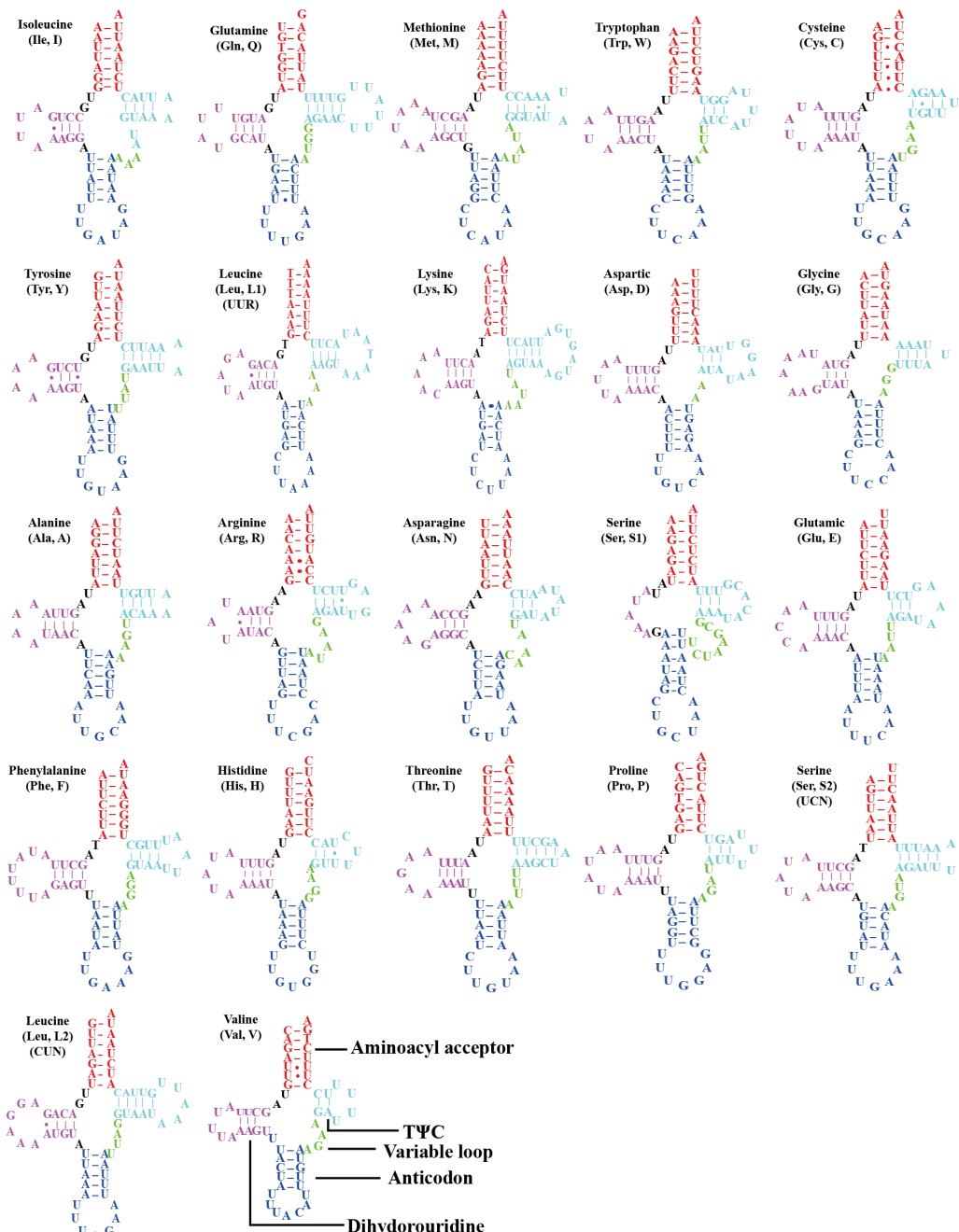

**Figure 3 Secondary structures of tRNAs in the mitogenome of Olidiana *ritcheriina*.** The dashes indicate Watson-Crick bonds and GU pairs, solid dots indicate mismatches.

Fig. S1D: P12-rR) had reached saturation (all *Iss <Iss. cSym* or *Iss. cAsym*, $p = 0.0000$) (Table 4; Fig. S1), thereby suggesting that the concatenated data is suitable for phylogenetic analysis.

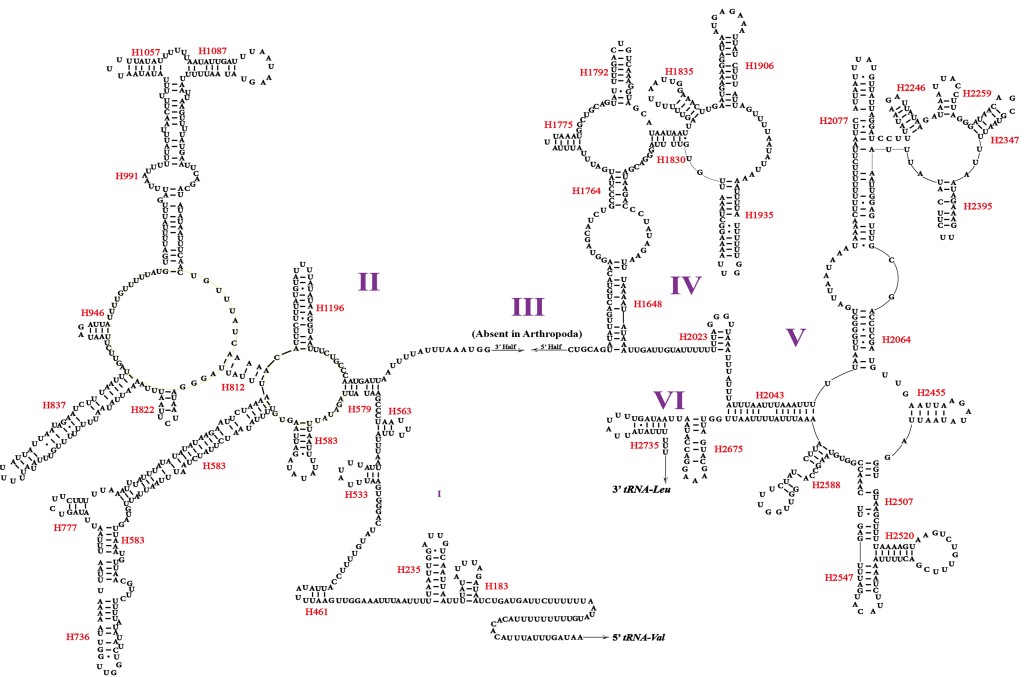

**Figure 4** **Predicted secondary structure of the *rrnL.* in the mitogenome of *Olidiana ritcheriina.*** Roman numerals indicate the conserved domain structure. Watson-Crick pairs are joined by dashes, hereas GU pairs are connected by dots.

All the 10 trees are presented in Fig. 7 and Fig. S2A–F. Almost all nodes received high support (posterior probability, PP >0.88) in BI analyses, whereas a few nodes received only moderate or low support in ML analyses of some datasets (bootstrap support, BS <75). Monophyly at the subfamily level within Membracoidea was strongly supported in all the trees. Membracidae as a sister group to Cicadellidae was well supported by all the results (*PP* > 0.94, BS = 100). Within Cicadellidae, the 37 species sampled in this study represent seven subfamilies and the main topology was as follows: (Deltocephalinae + ((Coelidiinae + Iassinae) + ((Typhlocybinae + Cicadellinae) + (Idiocerinae + (Treehopper + Megophthalminae))))) (Fig. 7). The results of BI and ML analyses generated results that are consistent with those of previous phylogenetic studies on the basis of combined morphological and molecular data (*Dietrich et al., 2001*; *Dietrich et al., 2017*; *Cryan et al., 2000*; *Cryan & Urban, 2012*; *Krishnankutty, 2013*; *Wang, Dietrich & Zhang, 2017*).

## DISCUSSION

The phylogenetic relationships inferred according to the five datasets showed slightly different topologies. In the BI-P123-rR/ML-P12-rR/ML-P123-rR analysis, the main topology was as follows: (Typhlocybinae + (Cicadellinae + (Deltocephalinae + ((Coelidiinae + Iassinae) + (Idiocerinae + (Treehopper + Megophthalminae)))))) (Fig. S2C). This topology is consistent with that reported in a previous study (*Du et al., 2017*) based on BI

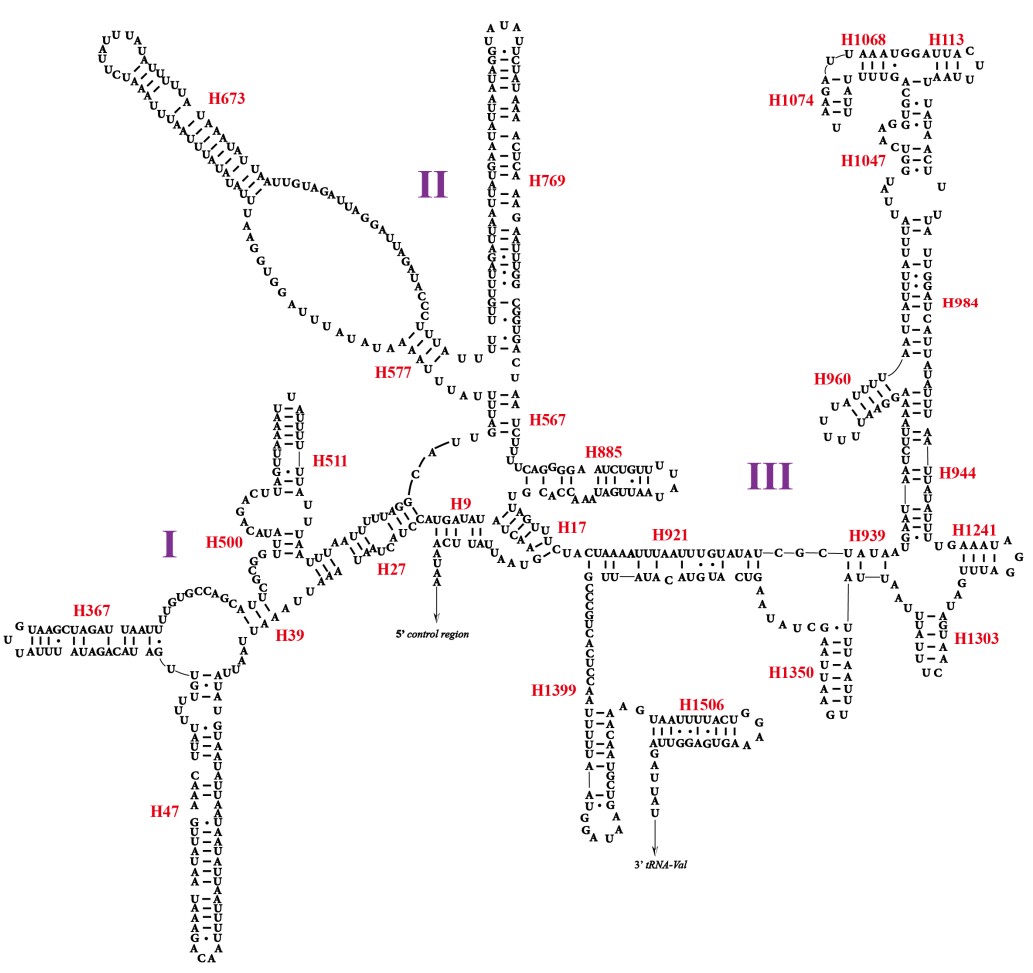

**Figure 5** **Predicted secondary structure of the *rrnS.* in the mitogenome of *Olidiana ritcheriina.*** Roman numerals indicate the conserved domain structure. Watson-Crick pairs are joined by dashes, hreas GU pairs are connected by dots.



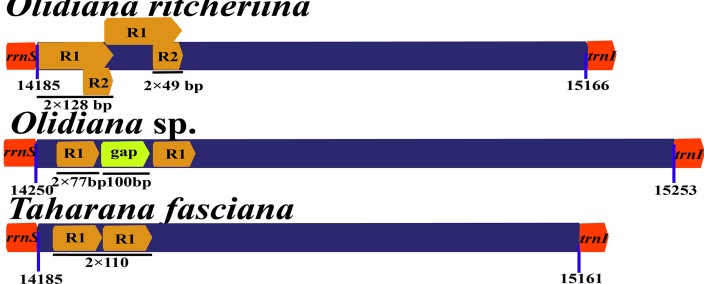

**Figure 6** **Organization of the control region structure in the mitochondrial genomes of *Olidiana ritcheriina.*** R, repeat unit.


**Table 4   Substitution saturation tests for the four dataset.**

| Dataset | Observed Iss | Iss.cSym[a] | Psym[b] | Iss.cAsym[c] | Pasym[d] | Dataset | Observed Iss | Iss.cSym[a] | Psym[b] | Iss.cAsym[c] |
|---|---|---|---|---|---|---|---|---|---|---|
| P123 | 0.419 | 0.817 | 0.0000 | 0.571 | 0.0000 | P123-rR | 0.420 | 0.818 | 0.0000 | 0.572 |
| P12 | 0.296 | 0.814 | 0.0000 | 0.570 | 0.0000 | P12-rR | 0.320 | 0.816 | 0.0000 | 0.571 |

**Notes.**

NumOUT = 32

[a]Critical values assuming a symmetrical tree.

[b]Signifcant difference between Iss and Iss.cSym (two-tailed test).

[c]Critical values assuming an extreme asymmetrical tree.

[d]Signifcant difference between Iss and Iss.cAsym (two-tailed $t$-test).

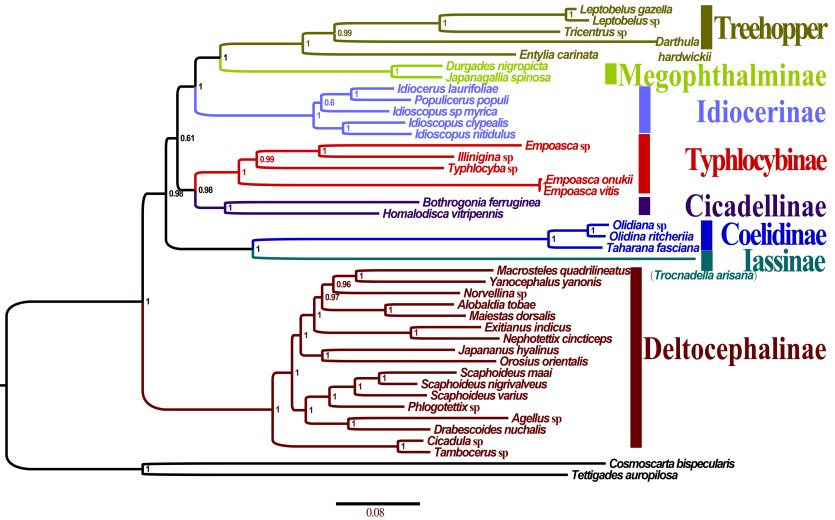

**Figure 7   Phylogenetic trees of *Olidiana ritcheriina*.** inferred based on the first and second codon positions of 13 PCGs using GTR+I+G model in MrBayes.

analysis of amino acid sequences. However, in some other studies (*Du et al., 2017*; *Wang et al., 2018*), the main topology has been reported to be different, i.e., (Deltocephalinae + (Typhlocybinae + (Cicadellinae + ((Coelidiinae + Iassinae) + (Idiocerinae + (Treehopper + Megophthalminae)))))) (Fig. S2D). This difference can be mainly attributed to the unstable positions of Deltocephalinae, Typhlocybinae, and Coelidiinae and Iassinae (Fig. 7).

In Membracoidea, three clades exhibited a stable sister relationship as shown in all trees in the present analysis: Idiocerinae + (Typhlocybinae + Cicadellinae); Coelidiinae + Iassinae; and Treehopper + Megophthalminae. This result is consistent with that reported in some previous studies (*Dietrich et al., 2001*; *Dietrich et al., 2017*; *Krishnankutty, 2013*; *Wang et al., 2017*; *Wang et al., 2018*). Coelidiinae was the most closely related to Iassinae in the present study according to the BI (*PP* = 1.00) and ML (*BS* = 100) trees, which were the same as those reported in previous studies (*Wang et al., 2017*; *Wang et al., 2018*). Within Coelidiinae, the three species sampled in the present study represent *Olidiana* and *Taharana*. The inferred relationships (*Taharana fascianus* + (*Olidiana* sp. + *Olidiana ritcheriina*)) were well supported by all BI (*PP* = 1.00) and ML (*BS* = 100) trees. The

third codon position shows higher saturation than the first and second codon positions (*Wei et al., 2010*; *Song, Liang & Bu, 2012*) (Table S2). Nevertheless, in our phylogenetic results, tree topologies were consistent regardless of whether the third codon position was excluded; however, this exclusion slightly increased support for some nodes in ML analyses (ML-13PCGs12/ML-13PCGs and ML-13PCGs12-2RNA/ML-13PCGs-2RNA) (Figs. S2C and S2F). The results of the present study are consistent with those of a previous phylogenetic study (*Du et al., 2017*).

## CONCLUSIONS

We sequenced the mitogenome of *O. ritcheriina* from Coelidiinae and presented their structure and sequence characteristics. Consistent with previous observations related to Membracoidea, the mitogenome of *O. ritcheriina* was highly conserved in terms of gene content, gene size, gene order, base composition, PCG codon usage, as well as tRNA and rRNA secondary structures.

Furthermore, the phylogeny of Membracoidea was inferred with all 40 complete mitogenomes, namely, 35 Cicadellidae and five Treehopper. The overall phylogenetic structure of Membracoidea is consistent with that reported in previous studies. Coelidiinae was grouped with a clade comprising Iassinae. The mitogenomic information of *O. ritcheriina* can be useful for future studies aimed at exploring the mitogenomic diversity of insects and evolution of related insect lineages.

The lack of complete mitogenomes of Coelidiinae sp. has restricted the understanding of the evolution of this group at the genome level. Therefore, further studies are required to elucidate the phylogenetic status of species belonging to this group and their relationships. In this context, the addition of more taxa and genes to the leafhopper mitogenomic dataset may contribute to the determination of the relationships shared among major leafhopper lineages.

### Funding
This study was supported by the National Natural Science Foundation of China (No. 31672342). The funders had no role in study design, data collection and analysis, decision to publish, or preparation of the manuscript.

### Grant Disclosures
The following grant information was disclosed by the authors:
National Natural Science Foundation of China: 31672342.

### Competing Interests
Zhi-Hua Fan is employed by Jingtanggang Customs House.
## Author Contributions

- Xian-Yi Wang and Ren-Huai Dai conceived and designed the experiments, performed the experiments, analyzed the data, contributed reagents/materials/analysis tools, prepared figures and/or tables, authored or reviewed drafts of the paper, approved the final draft.
- Jia-Jia Wang conceived and designed the experiments, analyzed the data, contributed reagents/materials/analysis tools, prepared figures and/or tables, authored or reviewed drafts of the paper, approved the final draft.
- Zhi-Hua Fan performed the experiments, analyzed the data, contributed reagents/materials/analysis tools, prepared figures and/or tables, approved the final draft.

## Field Study Permissions

The following information was supplied relating to field study approvals (i.e., approving body and any reference numbers):

The Institute of Entomology of Guizhou University approved field collection.

## DNA Deposition

The following information was supplied regarding the deposition of DNA sequences:

The species sequences are available at GenBank: MK738125.

## Data Availability

The mitogenome sequence of *Olidiana ritcheriina* is available as a Supplemental File.

## Supplemental Information

Supplemental information for this article can be found online at http://dx.doi.org/10.7717/peerj.8072#supplemental-information.

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
