# Peer review of "Complete mitogenome of Olidiana ritcheriina (Hemiptera: Cicadellidae) and phylogeny of Cicadellidae"

_PeerJ, doi:10.7717/peerj.8072_

## Round 0.1 · original submission · Major Revisions

· Academic Editor

Major Revisions

Dear Dr. Dai,
Thank you for your submission to PeerJ.
I agree with the three reviewers' suggestions. Especially, Reviewer 2's comments will help to improve your paper.
The methods need to be clearer. Did you generate an NGS template, and then confirm this using PCR and Sangon sequencing? How did you choose the model in the phylogenetic analysis?
You can consider testing the AGG-S and AGG-K as an issue according to the reviewer's suggestion.
I am looking forward to your revision.

With kind regards,
Jia-Yong Zhang
Academic Editor, PeerJ

Reviewer 1 ·

Basic reporting

English writing needs to be improved greatly.

Experimental design

Sequencing strategy in this paper is ambiguous. In Abstract, the author stated that mitogenomies were sequenced by NGS. However, in the section of Materials and methods, PCR and sanger sequencing were employed. It is unclear which of the sequencing method is used in this study.

Validity of the findings

2. One of the most important significance of mitogenome sequencing lies in providing additional molecular data for elucidating phylogenetic relationships of insects. What is the important issues required to be addressed in the Cicadellidae? This manuscript did not present any background or reviews on this point in the sections of Introduction or Results and Discussion. The results of phylogenetic analyses are too simple to make any sense on the phylogeny of Cicadellidae. Detailed phylogenetic results and associated discussion need to be provided.

Additional comments

P5 Line 33 srRNA > rRNAs.
P5 Lines 37-38 The description of relationships among subfamilies is insufficient by using only a set of parentheses.
P6 Lines 56-59 need to be rewrite for clarity.
P9 Line 184 amino acid codons > nucleotide positions

Reviewer 2 ·

Basic reporting

The article itself is well written, except for a few formatting errors, discussed below;

1) Methods: Line 94 – this section needs to be clarified. Did they generate an NGS template, and then confirm this my Sanger / PCR? I believe this to be the case, but the description is not that clear. Also – information on the data - # reads per base, etc are all needed at some point (a supplemental?) so readers can assess what was done more easily.

2) Figure 3b. In addition to the AGG-K issues discussed in “Validity of Findings", I am not sure that the RSCU data is best presented as a bar graph. Especially one where the bar heights are different. I would suggest either re-organizing the data so there is the 6-codon, then 4-codon, etc groups clustered on the X axis, or replacing this with a table of some kind. In addition – Ser2 has 4 codons, but the RSCU bar is about 5.7 in height (not 4), and Ser1 is between 2 and 3, while it should be either 4 (with the AAG-S error) or 3 (after correction). There is an error here, somewhere.

To the 4 points by the editor instructions, the presentation is clear, sufficiently references, other than the points above the structure and presentation is clear, and the article is self-contained with relevant results and discussion.

Experimental design

Other than the lack of clarity of the description of the sequencing methods and order described above, it is original research (but a standard analysis of a mt-genome). The research is well defined but a somewhat routine analysis, rigorous (if my assumption of sequencing is correct; again see above). Again, see "Basic Reporting" on how to improve the methods description of the sequence generation.

Validity of the findings

The most important analysis error is understandable, as this important observation has generally been ignored in arthropod mitogenome analyses. And this paper is 13 years old, so the authors may have missed it. Abascal et al. (PLoS Biology 2006, 4(5): e127.) showed that the mitochondrial genetic code often shifts within the arthropods, where AGG can encode Ser in the standard invertebrate code (I will abbreviate as AGG-S), but Lys in a modified variant of this code (AGG-K). The code even shifts within the Hemiptera, with Triatoma (Cimicomorpha), Homalodisca (Cicadellidae) and Philaenus (Cercopoidea) using the AGG-K code while other use the AGG-S. Thus, the species sequences, and those used in the phylogenic analysis were already predicted to be AGG-K.

The diagnostic for this shift appears to be the alterations of the Lys tRNA having a CUU anticodon paired with a GCU Ser-1 aniticodon. Figure 3 show both of these diagnostic anticodons. Thus the manuscript should point out that the AGG-K code is most likely used in this species (and most likely all those used in the phylogeny). I feel it is very important that this is incorporated, so that the message that these alterations are there, and that the phylogenetic of mtDNA can be problematic if this changing of AGG is not take into account.

This then impacts on the codon number and RSCU data presented (Figure 2). Lys and Ser1 will be 3-codon families (Lys - AAG, AAA and AGG; Ser1 AGA AGC and AGU). And it might be worth seeing what effect this has on the phylogenetic predictions (depending on who in the tree on Figure 6 uses what code).

In addition, there are errors in Figure 3's reporting of the RSCU results, outlined in the " Basic reporting" section.

Additional comments

The authors supply a relatively standard analysis of a new mitochondrial genome from a species within the Coelidiinae. The authors use a mix of NGS and PCR-Sanger confirmation to generate the genome (but see below). They also supply standard genome and phylogenic analysis.

There are a few things that need clarification, and re-analysis undertaken in light of this. This paper also presents an opportunity to see if the AGG-S and AGG-K is an issue or not for phylogentic reconstruction. The authors need to check the codon use of all genomes incorporated in the study to see if all are AGG-K as predicted (using the tRNAs and checking the relevant alignments for S>R or R>S issues). If they are all AGG-K, widening the phylogeny to AGG-S Hemiptera and seeing if the analysis is altered by this.

Reviewer 3 ·

Basic reporting

The manuscript describes sequencing and characterisation of the whole mitochondrial genome of the leafhopper, Olidiana ritcheriina. Overall, the findings are of interest to scientists interested in systematics of this and closely related insects.

Experimental design

I do find the experimental approach overall adequate, but several questions should be answered.
1. Line 130 Why you chose MAFFT 7.310 using the Q-INS-i algorithm for rRNA genes alignment?
2. Line 145 Why you chose using the GTR + G + I model for Bayesian (BI) analysis? Did this model was inferred by model selecting software or by your own experience?
3. Line 244 Are there some similar phylogenetic relationships when you compared with previous results using other genetic markers (e.g. Nuclear genes)?
4. The Table1, Figure 7 in supplementary material is better.
5. Did you use site-heterogeneous mixture GTR+G+I model in MrBayes? GTR is a site-homogeneous model, and MrBayes cannot execute GTR model for AA dataset. How did you realize that?

Validity of the findings

However, one major concern relates to the novelty of this work. Previous studies from author and others have also indicated that Coelidiinae members were clustered into a clade. I am struggling to see the novelty of this manuscript.

---

## Round 0.2 · Minor Revisions

· Academic Editor

Minor Revisions

According to the reviewers' comments, the paper requires some additional minor revisions. Thanks.

Reviewer 1 ·

Basic reporting

This version has been improved. However, some minor revisions are needed. English writing is still poor, especially in the sections of Discussion and Conclusions.

Experimental design

no comment

Validity of the findings

no comment

Additional comments

P5 L26 and P7 L100: Sangon sequencing should be Sanger sequencing.
P6 L65: Olidiana McKamey McKamey should be normal typeface.
P6 L66-68: this sentence need to be rewriten.
P6 L69: "provide support for" should be "provide insights for".
P6 L76-77 "and, thus, provide a basis for further molecular research on related taxa of the Cicadellidae species." is redundant.
P7 L114 loci should be boundaries.
P8 L126-129 this sentence need to be rewriten for clearity.
P10 L208-209 Triatoma, Homalodisca and Philaenus are the names of genus, they shold be italic.
P11 L241 five concatenated nucleotide sequences should be five concatenated nucleotide sequence datasets.
P11 L245-248 saturation analysis address the issue on whether some positions or partitions of a dataset are saturated, and to test whether these sites can be used for futher phylogenetic analysis.
P12 L291 presenting should be presented.
P12 L292-293 "The assembled and annotated mitogenome has been submitted to the GenBank of the NCBI with the accession number MK738125.“ is redundant.
P12 L300-301 "The Coelidiinae was directly grouped into one clade with Iassinae lineages." >> The Coelidiinae grouped with a clade comprising Iassinae.
P12 L306 imperative >> needed
P12 L308 help in improving >> contribute to
P12 L209 delete "the still poorly understood".

"sp." in Figure 7 should be normal typeface.

Reviewer 2 ·

Basic reporting

Some of the corrected sections are unclear. Another re-reading is necessary.

Experimental design

Fine. But there are still math errors in the RSCU figure, implying a problem.

Validity of the findings

See Figure 2 problem, mentioned in 2.

Additional comments

Line 26-27 “Sequencing data were analyzed using maximum likelihood and Bayesian analyses.”. Do you mean “Phylogenic trees were constructed using maximum likelihood and Bayesian analyses.

207 -209 “Abascal et al. (2006; 2012) have showed that the code even shifts within…” should read “Abascal et al. (2006; 2012) have showed that the invertebrate mitochondrial genetic code even shifts within…”

Figure 2 – There are still errors in the RSCU graph. Ser2 sums to ~5.75 instead of 4, Ser1 and Cys should sum to 3 each.

Figure 3 – Labels A-V could just be removed. This time, I was confused what tRNA “B” was at a quick glance.

---

## Round 0.3 · Minor Revisions

· Academic Editor

Minor Revisions

Dear Dr. Dai,

Before a final decision is rendered, the Section Editors for this part of the journal have commented on your submission. They believe that further minor revision is needed.

Specifically, they said:

“I do not understand how the authors can claim that the sum of the RSCU does not add up to the number of codons.

I also think that the language in lines 206-201 is deficient. For the text:

"Abascal et al. (2006; 2012) have shown that the invertebrate mitochondrial genetic code even shifts within the Hemiptera, with Triatoma (Cimicomorpha), Homalodisca (Cicadellidae), and Philaenus (Cercopoidea) using the AGG codon that was translated as Lys instead of Ser; therefore, the AGG codon in O. ritcheriina was translated as Lys instead of Ser."

I suggest changing it to:

"Abascal et al. (2006; 2012) have shown that the invertebrate mitochondrial genetic code shifts within the Hemiptera, with Triatoma (Cimicomorpha), Homalodisca (Cicadellidae), and Philaenus (Cercopoidea) using the AGG codon that was translated as Lys instead of Ser; accordingly, our tRNA analysis shows that the AGG codon in O. ritcheriina is translated as Lys instead of Ser."

Please can you address these issues in a final revision round.

---

## Round 0.4 · accepted · Accept

· Academic Editor

Accept

I am writing to inform you that your manuscript - Complete mitogenome of Olidiana ritcheriina (Hemiptera: Cicadellidae) and phylogeny of Cicadellidae - has been Accepted for publication. Congratulations!